# REINFORCEMENT LEARNING FOR LARGE GROUP SYSTEMS USING HIERARCHICAL KERNEL REPRESENTATIONS

## ABSTRACT

Policy learning for targeted coordination of massive-scale populations of, in the limit a continuum spectrum of, intelligent agents is severely underexplored with sparse literature in reinforcement learning research. The purpose of this work is to fill in this gap by addressing the major challenge: the curse of dimensionality caused by the huge population size. To this end, we formulate such an intelligent agent population as a parameterized deterministic dynamical system defined on a function space, referred to as a group system. This in turn gives rise to a functional setting of reinforcement learning problems involving group systems. A novel moment transform is then proposed to give a kernel representation of group systems, under which we develop a hierarchical algorithm for learning optimal policies of group systems. As a significant advantage, each hierarchy of the algorithm preserves the optimality of all its lower-level children, which then leads to the fast convergence of the algorithm with a theoretical guarantee.

## 1 INTRODUCTION

Reinforcement learning (RL), a major machine learning paradigm, has been recognized as one of the most powerful tools for shaping the desired behavior of intelligent agents by learning optimal policies in an autonomous manner. This scope naturally identifies control theory as a primary discipline in which RL is extensively investigated, and prominent examples of RL for control systems range from path planning for robotic agents and cyber-attack detection for cyber-physical systems to safety validation for autonomous vehicles (Kober et al., 2013; Li & Qiu, 2019; Bertsekas, 2019; Dixon et al., 2020; Meyn, 2022). In this decade, the research stream of learning to control has rapidly shifted to large-scale populations of intelligent agents arising from diverse cutting-edge applications, including neural stimulation and brain medicine (Marks, 2005; Wilson, 2005; Shoeb, 2009; Bomela et al., 2020), quantum control and quantum machine learning (Glaser et al., 1998; Li et al., 2011; Dong et al., 2008; Chen et al., 2014), and complex networks and network inference (Zlotnik et al., 2016; Abel et al., 2016; Wang et al., 2018). In particular, the massive scale of such populations significantly increases the complexity of understanding their behavior toward the fundamental limit, e.g., a neural population in neuroscience may contain up to $\sim 10^{11}$ neuron cells (Zlotnik & Li, 2012; Li et al., 2013; Ching & Ritt, 2013), and a spin sample in quantum science generally consists of $\sim 10^{23}$ nuclear spins (Li & Khaneja, 2006; Kobzar et al., 2005). In addition, these populations generally only allow broadcasting control policies applied at the population level to guide each individual agent as desired, which is beyond the scope of RL algorithm developed for Markov decision processes. This in turn drives the urgent demand for a general theory inclusive for RL of such massive-scale group systems.

Driven by the hope to fill in this literature gap, this work is devoted to the development of a novel RL framework for learning control policies of group systems. The emphasis is on establishing the fundamentals that lead to an effective algorithmic approach targeted at those systems composed of a huge number, in the limit a continuum, of intelligent agents. This in turn sheds light on the generizability of the proposed framework in the sense of its applicability regardless of the size of group systems.

**Our contributions.** We propose a systematic formulation of populations of intelligent agents responding to a common policy in terms of deterministic group systems defined on function spaces, which in turn gives rise to the formulation of RL problems for group systems in a functional setting. Then, we introduce a moment kernel representation for group systems, under which we develop a hierarchical algorithm for RL of group systems. In particular, this hierarchical structure has the optimality-preserving property, leading to the fast convergence of the proposed algorithm with a theoretical guarantee. The contributions of our work can then be summarized as follows.

- Systematic formulations of populations of intelligent agents as deterministic group systems defined on function spaces and RL problems for group systems in the functional setting, regardless of the population size.
- Development of the moment kernel representation for group systems.
- Design of a hierarchical algorithm for RL of group systems with fast convergence that is verified both theoretically and numerically.

**Related works.** Reinforcement learning of large-scale populations of intelligent agents has been witnessed to attract increasing attention in recent years. In the deterministic dynamical systems setting, a current active research focus is on multi-agent systems by using distributed learning and optimization infrastructures (Menger) (Yazdanbakhsh et al., 2020), retrieval mechanisms (Humphreys et al., 2022), and deep neural network techniques (Chu et al., 2020; Andrychowicz et al., 2021; Fu et al., 2022; Sarang & Poullis, 2023), where the scale of the systems varies from hundreds to thousands. However, allowing the number of agents in a group system to approach infinity as considered in this work remains under-explored with sparse literature in the RL society. In the stochastic dynamical systems setting, RL of Markov decision processes has been a prosperous research area for decades, in which dynamic programming techniques serve as the main tool (van Otterlo & Wiering, 2012; García et al., 2015; Sutton & Barto, 2018; Bertsekas, 2019). Moreover, the formulation of RL problems over infinite-dimensional spaces has only been proposed in the stochastic setting, for the purpose of learning feedback control policies of stochastic partial differential equation systems by using variational optimization methods (Evans et al., 2019).

## 2 PROBLEM FORMULATION: REINFORCEMENT LEARNING OF GROUP SYSTEMS

Despite the prosperity of RL research for decades, policy learning for large-scale populations of intelligent agents remains challenging from both theoretical and practical perspectives. In addition to the huge size, the lack of a systematic mathematical formulation of such populations in the RL setting is also a major issue, which will be tackled in this section.

### 2.1 GROUP SYSTEM FORMULATION OF POPULATION AGENTS

Many large-scale population agents, e.g., nuclear spin samples, robot swarms, and neural ensembles, can be formulated as parameterized differential equation systems in the form of

$$\frac{d}{dt}x(t,\beta) = F(\beta, x(t,\beta), u(t)), \tag{1}$$

where the system parameter $\beta$ taking values on a compact set $\Omega \subset \mathbb{R}$, $x(t,\beta) \in \mathbb{R}^n$ is the state of the $\beta^{\text{th}}$ agent, and $u(t) \in \mathbb{R}^m$ the policy implemented by every agent at time $t$ (Li & Khaneja, 2006; 2009; Becker & Bretli, 2012; Li et al., 2013; Zlotnik & Li, 2012). Note that this group system formulation of population agents is regardless of the population size, equivalently the cardinality of the system parameter space $\Omega$, indicating its advantage of dealing with large-scale populations.

On the other hand, it is also worth mentioning that there is no randomness and dynamics involved in $\beta$, and hence the group system formulation in (1) is fundamentally different from the Markov decision process formulation commonly used in RL. This further stimulates the demand for a novel RL framework to learn control policies for such group systems. To initialize the development, we first note that with $\beta$ varying on $\Omega$, the state $x(t,\cdot)$ of the group system can be viewed as an $\mathbb{R}^n$-valued function defined on $\Omega$. In the limiting case that the group system is composed of infinitely many

intelligent agents, equivalently $\Omega$ is an infinite set, the space $\mathcal{F}(\Omega, \mathbb{R}^n)$ of such functions is infinite-dimensional, which rises to a significant challenge to the policy learning problem. Technically, for the purpose of developing an RL approach over the infinite-dimensional function space to tackle this learning problem, we further require $x(t, \cdot)$ to be square integrable, i.e., $\mathcal{F}(\Omega, \mathbb{R}^n) \subseteq L^2(\Omega, \mathbb{R}^n)$.

## 2.2 REINFORCEMENT LEARNING OVER FUNCTION SPACES

Given an immediate reward $r\big(x(t, \beta), u(t)\big)$, it is generally impossible to simultaneously minimize the (discounted) future rewards $V_\beta(x(t, \beta)) = \int_t^\infty e^{-\lambda s} r(x(s, \beta), u(s)) ds$ for all the agents by a universal policy $u(t)$ (see Appendix **??** for an example), where $\lambda > 0$ is the discount factor. Instead, we define the value function of the group system in (1) as the "averaged further rewards" over all the agents

$$V(x_t) = \int_\Omega V_\beta(x(t, \beta)) d\beta = \int_\Omega \int_t^\infty e^{-\lambda s} r(x(s, \beta), u(s)) ds d\beta, \qquad (2)$$

where $x_t(\cdot) = x(t, \cdot)$ denotes the state of the group system in (1) as a function defined on $\Omega$. However, to guarantee the existence of the optimal policy $u^*(t)$ such that $V^*(x_t) = \int_\Omega \int_t^\infty e^{-\lambda s} r(x^*(s, \beta), u^*(s)) ds d\beta = \min_{u|_{[t,\infty)}} \int_\Omega \int_t^\infty e^{-\lambda s} r(x(s, \beta), u(s)) ds d\beta$, more stringent conditions on the agent dynamics and immediate reward function than classical RL problems need to be imposed, where $x^*(t, \beta)$ is the optimal trajectory and $u_{[t,\infty)]}$ denotes the restriction of the policy, as an $\mathbb{R}^m$-valued function defined on $[0, \infty)$, on $[t, \infty)$.

**Assumption S1.** The policy $u : [0, T] \to \mathbb{R}^m$ takes values on a compact subset $U \subset \mathbb{R}^m$.

**Assumption S2.** The function $F : \Omega \times \mathbb{R}^n \times U \to \mathbb{R}^n$ is bounded and Lipschitz continuous in the state variable uniformly, that is, there is a constant $L$ such that $|F(\beta, x, a) - F(\beta, y, a)| \le L|x - y|$ for all $x, y \in \mathbb{R}^n$, $\beta \in \Omega$, and $a \in U$, where $|\cdot|$ denotes a norm on $\mathbb{R}^n$.

Assumptions 1 and 2 guarantee that, with any policy implemented, the system representing each agent has a unique and Lipschitz continuous solution for almost every $t \ge 0$ (Arnold, 1978).

**Assumption C1.** The immediate reward $r : \mathbb{R}^n \times U \to \mathbb{R}$ is bounded and Lipschitz continuous in the state variable uniformly.

**Assumption C2.** The discounted immediate reward is integrable for any policy, i.e., $\int_\Omega \int_0^\infty e^{-\lambda t} |r(x(t, \beta), u(t))| dt d\beta < \infty$.

The above assumptions not only warrant that the optimal value function $V^*$ is well-defined but also make it satisfy some regularity conditions.

**Proposition 1 (Regularity of the optimal value function)** *Given a group system as in (1) satisfying Assumptions S1 and S2 with an immediate reward satisfying Assumptions C1 and C2. Then, an optimal policy $u^*$ exists. In addition, if the discount factor $\lambda > L$, then the optimal value function $V^*$ is bounded and Lipschitz continuous; otherwise, $V^*$ is Hölder continuous with some exponent $0 < \alpha < 1$, where $L$ denotes the Lipschitz constant of $F$.*

*Proof.* For the existence of $u^*$, we note that the value function $V$ is also a function of $u$. Then, it suffices to show that $V$ is a continuous function of $u$ and the space of all control polices satisfying Assumption S1 is compact, because any continuous function defined on a compact space attains its minimum. The regularity of $V^*$ directly follows from the dynamic programming principle. See Appendix A for the detailed proof. □

## 3 HIERARCHICAL REPRESENTATION FOR REINFORCEMENT LEARNING OF GROUP SYSTEMS

The existence of optimal policies discussed in the previous section lays the foundation for the development of an RL approach to training the dynamic behavior of group systems consisting of

large-scale intelligent agents. One major challenge to this end is unarguably the large group size, leading to the high dimension of group systems, so that the learning problem experiences the curse of dimensionality. To address this challenge, we will propose the novel moment representation of group systems, which gives rise to a hierarchical structure of the RL problem for tackling it in a low-dimensional environment.

## 3.1 MOMENT KERNEL REPRESENTATIONS OF GROUP SYSTEMS ON FUNCTION SPACES

The method of moments concerns with representing a function or probability distribution in terms of a sequence of numbers, called the moments. It was established by the Russian mathematicians P. L. Chebyshev and A. Markov at the end of 19 century, and then extensively studied under different settings, notably the Hamburger, Hausdorff, and Stieltjes moment problems (Hamburger, 1920; 1921; Hausdorff, 1923; Stieltjes, 1993). The most general formulation of the moment problem in the modern language was proposed by the Japanese mathematician K. Yosida (Yosida, 1980). The functional interpretation of group systems mentioned in Section 2.1 opens up the possibility of introducing the method of moments to study group systems.

To fix the idea, we consider the case that the group system in (1) evolves on a Banach space $\mathcal{B}$ consisting of real-valued integrable functions. We further assume that the dual space $\mathcal{B}^*$, i.e., the space of continuous linear functionals on $\mathcal{B}$, is separable, and hence has a countable basis $\{\Phi_k\}_{k\in\mathbb{N}}$. Then, we define the $k^{\text{th}}$-moment of the group system to be

$$m_k(t) = \langle \Phi_k, x_t \rangle, \tag{3}$$

where $\langle \cdot, \cdot \rangle : \mathcal{B}^* \times \mathcal{B} \to \mathbb{R}$ denotes the dual-primal space pairing, i.e., $m_k(t)$ is essentially the evaluation of the function $\Phi_k : \mathcal{B} \to \mathbb{R}$ at the point $x_t \in \mathcal{B}$. To be more specific, because $x_t$ is assumed to be integrable, $\Phi_k$ can be represented by an integral kernel such that $m_k(t) = \langle \Phi_k, x_t \rangle = \int_\Omega \Phi_k x_t d\mu$ for some measure $\mu$ on $\Omega$. On the other hand, Yosida's moment problem implies that the state $x_t$ of the group system corresponds to the associated *moment sequence* $m(t) = (m_0(t), m_1(t), \dots)$ in a one-to-one manner and vice versa (Yosida, 1980). Indeed, we can think of $m(t)$ as the kernel representation of the group state $x_t$ with respect to these moment kernels $\Phi_k, k \in \mathbb{N}$.

The moment kernel representation of the group system in (1) can be derived by using the linearity of the dual-primal pairing as $\frac{d}{dt}m_k(t) = \frac{d}{dt}\langle \Phi_k, x_t \rangle = \langle \Phi_k, \frac{d}{dt}x_t \rangle = \langle \Phi_k, F(\cdot, x_t, u(t)) \rangle$, where the second equality follows from the continuity of the functional $\Phi_k$. A comparison between the term $\langle \Phi_k, F(\cdot, x_t, u(t)) \rangle$ and the definition of the moment in (3) immediately reveals that it is nothing but the $k^{\text{th}}$-moment $F(\beta, x_t, u(t))$ as a function of $\beta$. Let $\bar{F}$ denote the entire moment sequence of $F$, then we obtain the moment kernel representation of the group system as

$$\frac{d}{dt}m(t) = \bar{F}(m(t), u(t)). \tag{4}$$

The derivation of the *moment system* in (4) further indicates that the moment kernels $\Phi_k$ not only kernelize the group state as in (3) but also the entire group system.

More importantly, it is worth noticing that even for a group system composed of a continuum of intelligent agents, i.e., $\Omega$ is an uncountable set, its moment kernel representation always contains countably many components. This means that the moment kernelization process can be generically used as a model reduction to massive-scale group systems, which is also the key feature leading to the desired hierarchical structure of RL problems for group systems.

## 3.2 HIERARCHICAL ALGORITHM FOR REINFORCEMENT LEARNING OF MOMENT KERNELIZED GROUP SYSTEMS

Of course, to carry over RL of group systems to the moment domain, it is inevitable to find the value function $V$, defined in (2), in moment kernel representation as well. The derivation is similar to that of the moment system. Specifically, the integrability assumption on the immediate reward $r$ (Assumption C2) implies that the order of the two integrals in $V$ can be changed, yielding $V(x_t) = \int_t^\infty e^{-\lambda s} \left[ \int_\Omega r(x(s, \beta), u(s)) d\beta \right] ds$, in which the integral with respect to $\beta$ essentially calculates the $0^{\text{th}}$-moment of $r(x(s, \beta), u(s))$ as a function of $\beta$. Denoting this quantity by $\bar{r}(m(t), u(t))$, we obtained the moment kernelized value function as $V(m(t)) = \int_t^\infty e^{-\lambda s} \bar{r}(m(s), u(s)) ds$.

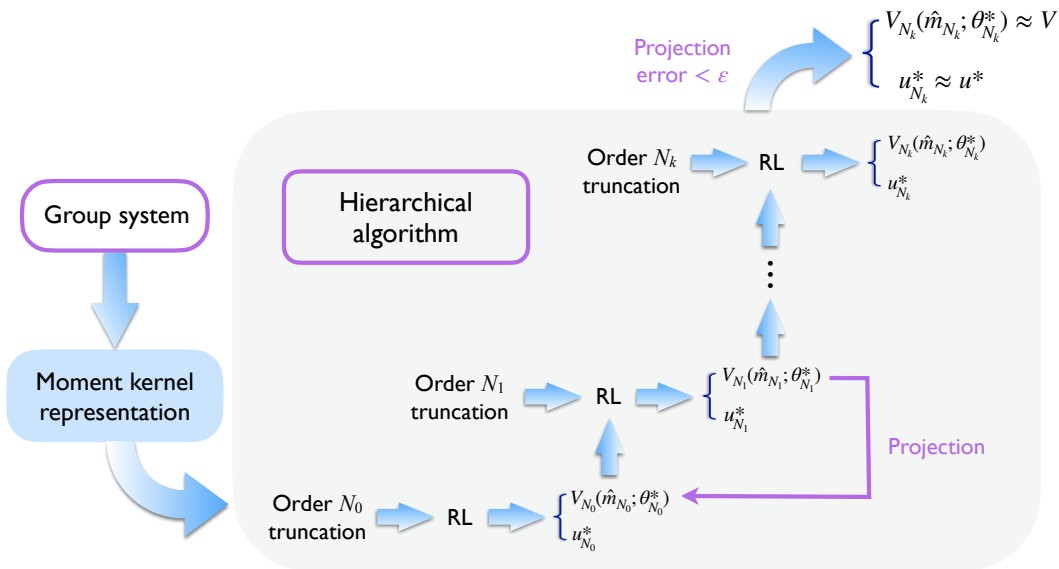

Figure 1: Workflow of the proposed hierarchical algorithm for RL of group systems in the moment kernel representation.

Owing to the aforementioned countable nature of the moment sequence $m(t) = (m_0(t), m_1(t), \dots)$, we consider the hierarchy formed by the truncation of the moment kernelized RL problem. To elaborate it, we let $\widehat{m}_N(t) = (m_0(t), m_1(t), \dots, m_N(t))$ be the truncated moment sequence up to order $N$, and $V_N(\widehat{m}_N(t))$ be the value function associated to the order $N$ truncated moment system, that is, $\frac{d}{dt}\widehat{m}_N(t) = \widehat{F}_N(\widehat{m}_N(t), u(t))$, where as before $\widehat{F}_N$ is the truncation of $\bar{F}$ up to order $N$. It is crucial to observe that $V_N$ is essentially the restriction of $V$ on the space $\mathcal{M}_N$ consisting of all order $N$ truncated moment sequences. Therefore, $V$ can be approximated by $V_N$ for large enough $N$, which can then be algorithmically realized by solving a sequence of RL problems with the truncation order $N$ increasing as shown in Figure 1.

Specifically, we parameterize the value function $V$ of the moment kernerlized group system by a parameter vector $\theta \in \mathbb{R}^r$ and represent it as $V(m;\theta)$, where $\theta$ can be associated with a neural network, a kernel function or simply a regression slope (Bertsekas, 2019; Meyn, 2022). Starting with an initial truncation order $N_0$ and parameter value $\theta_{N_0}^{(0)}$, the best approximation $V_{N_0}(\widehat{m}_{N_0}, \theta_{N_0}^*)$ of $V_{N_0}^*(\widehat{m}_{N_0})$, the optimal value function for the order $N_0$-truncated moment system, can be learned by solving $\theta_{N_0}^* = \text{argmin}_\theta \mathcal{L}(D_{N_0}(t;\theta))$, where $\mathcal{L}$ is a loss function, e.g., the $L^2$-error $\mathcal{L}(D_{N_0}(t;\theta)) = \int_0^\infty |D_{N_0}(t;\theta)|^2 dt$, and $D_{N_0}(t;\theta) = V_{N_0}(\widehat{m}_{N_0}^*(t);\theta) - \int_0^t e^{-\lambda t}\bar{r}(\widehat{m}_{N_0}^*(s), \widehat{u}_{N_0}^*(s))ds + e^{-\lambda t}V_{N_0}(\widehat{m}_{N_0}^*(t);\theta)$ is the temporal difference. In particular, the optimal policy $u_{N_0}^*$ can be obtained by minimizing the system Hamiltonian as $u_{N_0}^*(t) = \text{argmin}_a H\big(\widehat{m}_{N_0}^*(t), \nabla V_{N_0}(\widehat{m}_N^*(t)), a\big) = \text{argmin}_a\big\{r(\widehat{m}_{N_0}^*(t), a) + \langle\nabla V_{N_0}(\widehat{m}_{N_0}^*(t)), \widehat{F}_{N_0}(\widehat{m}_{N_0}^*(t), a\rangle\big\}$, and is generally represented in the feedback form as a function of $\widehat{m}_{N_0}^*(t)$. Then, the optimization $\min_\theta \mathcal{L}(D_{N_0}(t;\theta))$ can be solved by a standard value iteration or policy improvement (Doya, 2000). Denoting $\theta_{N_0}^* = \text{argmin}_\theta \mathcal{L}(D_{N_0}(\theta))$, we then pick a new truncation order $N_1 > N_0$ and learn $V_{N_1}$ by using the same procedure from the initial condition $\theta_{N_1}^{(0)} = \theta_{N_0}^*$. Keep increasing the truncation order and repeat the above learning procedure until a truncation order $N_k$ is reached so that the *projection error* (see Remark 1 below for the explanation of this terminology) $P_{N_k}(\widehat{m}_{N_k}^*(t)) = V_{N_k}(\widehat{m}_{N_k}^*(t);\theta_{N_k}^*) - V_{N_{k-1}}(\widehat{m}_{N_k}^*(t);\theta_{N_{k-1}}^*)$ satisfies $\mathcal{L}(P_{N_k}(\widehat{m}_{N_k}^*(t))) < \varepsilon$ for some predetermined approximation accuracy $\varepsilon > 0$. This algorithm is shown in Algorithm 1.

**Theorem 1 (Moment convergence of reinforcement leaning for group systems)** *Let $u_N^*$ and $V_N^*$ denote the optimal policy and optimal value function of the order $N$ truncated moment system, respectively. Then, there is a sequence of truncation orders $N_i \in \mathbb{N}$ such that $u_{N_i}^* \to u^*$ and*

---

**Algorithm 1** Hierarchical algorithm for RL of group systems

---

**Input:** $x_0, F, r, \lambda, \mathcal{L}, \varepsilon$
**Output:** $u^*$

    *Initialization* : Given $N_0, \theta_{N_0}^{(0)}, P, j = 0$
2: Compute $\bar{r}$
    **while** $P \geq \varepsilon$ **do**
4:     Compute $\widehat{m}_{N_j}(0), \widehat{F}_{N_j}$
       Solve    $u_{N_j}^*(t)$    =    $u_{N_j}^*(\widehat{m}_{N_j}^*(t))$    =    $\mathrm{argmin}_a \{\bar{r}(\widehat{m}_{N_j}^*(t), a)$  +
      $\langle \nabla V_{N_j}(\widehat{m}_{N_j}^*(t); \theta), \widehat{F}_{N_j}(\widehat{m}_{N_j}^*(t), a) \rangle\}$
6:     Solve $\theta_{N_j}^* = \mathrm{argmin}_\theta \mathcal{L}(D_{N_j}(t; \theta))$ with the initial condition $\theta_{N_j}^{(0)}$ and $D_{N_j}(t; \theta) =$
      $V_{N_j}(\widehat{m}_{N_j}^*(t); \theta) - \int_0^t e^{-\lambda t} \bar{r}(\widehat{m}_{N_j}^*(s), \widehat{u}_{N_j}^*(s)) ds + e^{-\lambda t} V_{N_j}(\widehat{m}_{N_j}^*(t); \theta)$
      $P_{N_j}(\widehat{m}_{N_j}^*(t)) = V_{N_j}(\widehat{m}_{N_k}^*(t); \theta_{N_j}^*) - V_{N_{j-1}}(\widehat{m}_{N_j}^*(t); \theta_{N_{j-1}}^*)$
8:     $P = \mathcal{L}(P_{N_j}(\widehat{m}_{N_j}^*(t)))$
      $j \leftarrow j + 1$
10:    $\theta_j^{(0)} = \theta_{N_{j-1}}^*$
      Pick $N_j > N_{j-1}$
12: **end while**
    $u^*(t) = \mathrm{argmin}_a \{\bar{r}(\widehat{m}_{N_{j-1}}^*(t), a) + \langle \nabla V_{N_{j-1}}(\widehat{m}_{N_{j-1}}^*(t); \theta_{N_{j-1}}^*), \widehat{F}_{N_{j-1}}(\widehat{m}_{j-1}^*(t), a) \rangle\}$
14: **return** $u^*$

---

*$V_{N_i}^* \to V^*$ as $i \to \infty$, where $u^*$ and $V^*$ are the optimal policy and optimal value function of the moment system in* [4].

*Proof.* The first step is to show the convergence of $V_{N_i}^*$ to $V^*$ by fully utilizing the fact that $V_{N_i}^*$ is bounded by the restriction of $V^*$ on the space consisting of order $N_i$ truncated moment sequences. The second step is to use the first step to show that $V^*$ is the viscosity solution of a Hamilton-Jacobi-Bellman equation. See Appendix B for the detailed proof. $\square$

Theorem 1 provides the convergence proof of Algorithm 1, and also offers the theoretical guarantee that the limiting policy and value function are exactly the optimal policy and value function of the moment kernelized, and hence the original, group system. Before demonstrating the applicability of the learning algorithm by using some examples, we would like to point out its remarkable feature due to the hierarchical structure.

**Remark 1 (Optimaliy preserving hierarchy)** *By the definition of the value function for the truncated moment system, for any $j > i$, $V_{N_i}$ is essentially the restriction / projection of $V_{N_j}$ on $\mathcal{M}_{N_i}$. In the algorithm, because $V_{N_j}(\cdot; \theta_{N_j}^*)$ gives the best approximation of $V_{N_j}^*$, its restriction on $\mathcal{M}_{N_i}$ necessarily approximates $V_{N_i}^*$ at least as good as $V_{N_i}(\cdot; \theta_{N_j}^*)$ does. Therefore, this hierarchical learning structure, with respect to the increasing of the truncation order, indeed preserves the optimality in each hierarchy.*

*Conversely, the hierarchical structure also guarantees that higher-ranking problems always start with better initial conditions, the minimizers of their children problems. Together with the fact that higher-ranking problems have higher dimensions, it further demonstrates the high efficiency of the proposed algorithm.*

## 4  EXAMPLE AND SIMULATION

To demonstrate the applicability as well as the effectiveness and efficiency of the proposed RL algorithm, in this section, we will conduct a detailed investigation into the linear quadratic regulator (LQR) problem for group systems. It is also worth mentioning that although the LQR problem for classical linear systems has been thoroughly studied, it remains unexplored for group systems.

To fix the idea, we consider the group system $\frac{d}{dt}x(t, \beta) = \beta x(t, \beta) + u(t)$ defined on $L^2([-1, 1])$, the space of real-valued square integrable functions defined on $[-1, 1]$, with the immediate reward

and discount factor given by $r(x(t, \beta), u(t)) = x^2(t, \beta) + \frac{1}{2}u^2(t)$ and $\lambda = -2.5$, respectively. Then, the value function is defined as $V(x_t) = \int_{-1}^{1} \int_t^\infty e^{-2.5t} \left[ x^2(t, \beta) + \frac{1}{2}u^2(t) \right] dt d\beta$.

**Moment kernel representation.** The initial step in Algorithm 1 is to kernelize the group system and value function in terms of moment sequences. In particular, we choose the moment kernels $\{\Phi_k\}_{k \in \mathbb{N}}$ to be the set of Chebyshev polynomials, which is an orthonormal basis of $L^2([-1, 1])$ and satisfies the recursive relation $2\beta\Phi_k(\beta) = \Phi_{k-1}(\beta) + \Phi_{k+1}(\beta)$ for all $k \in \mathbb{N}$, where we defined $\Phi_{-1} = 0$. Then, this recursive relation can be applied to derive the moment kernel representation of the system as $\frac{d}{dt}m_k(t) = \frac{d}{dt}\int_{-1}^{1}\Phi_k(\beta)x(t, \beta)d\beta = \int_{-1}^{1}\Phi_k(\beta)\frac{d}{dt}x(t, \beta)d\beta = \int_{-1}^{1}\Phi_k(\beta)[\beta x(t, \beta) + u(t)]d\beta = \int_{-1}^{1}\beta\Phi_k(\beta)x(t, \beta)d\beta + \int_{-1}^{1}\Phi_k u(t)d\beta = \frac{1}{2}\int_{-1}^{1}[\Phi_{k-1}(\beta) + \Phi_{k+1}(\beta)]x(t, \beta)d\beta + \int_{-1}^{1}\Phi_k u(t)d\beta$, giving $\frac{d}{dt}m_0(t) = \frac{1}{2}m_1(t) + u(t)$ and $\frac{d}{dt}m_k(t) = \frac{1}{2}[m_{k-1}(t) + m_{k+1}(t)]$ for $k \geq 1$. Next, by using the $L^2$-orthonormality of $\{\Phi_k\}_{k \in \mathbb{N}}$, we obtain the moment kernelized value function as $V(m(t)) = \int_{-1}^{1} \int_t^\infty e^{-2.5t} \left[ x^2(t, \beta) + \frac{1}{2}u^2(t) \right] dt d\beta = \int_t^\infty e^{-2.5t} \int_{-1}^{1} \left[ x^2(t, \beta) + \frac{1}{2}u^2(t) \right] d\beta dt = \int_t^\infty e^{-2.5t}[\|m(t)\|^2 + u^2(t)]dt$, where $\|m(t)\| = [\sum_{k=0}^\infty m_k^2(t)]^{1/2}$ is the $\ell^2$-norm of $m(t)$. This further indicates $m(t) \in \ell^2$, the space of square summable sequences so that the moment kernerlized system is a dynamical system defined on $\ell^2$.

**Moment truncation.** For any truncation order $N$, the truncated moment system is given by a linear system defined on $\mathbb{R}^N$ in the form of

$$\frac{d}{dt}\widehat{m}_N(t) = \widehat{A}_N\widehat{m}_N(t) + \widehat{B}_N u(t)$$

$$= \begin{bmatrix} 0 & 1 & 0 & \cdots & 0 & 0 \\ 1 & 0 & 1 & \cdots & 0 & 0 \\ 0 & 1 & 0 & \cdots & 0 & 0 \\ & & & \ddots & & \\ 0 & 0 & 0 & \cdots & 0 & 1 \\ 0 & 0 & 0 & \cdots & 1 & 0 \end{bmatrix} \widehat{m}_N(t) + \begin{bmatrix} 1 \\ 0 \\ 0 \\ \vdots \\ 0 \\ 0 \end{bmatrix} u(t),$$

and the value function restricted to the space of order $N$ truncated moment sequences is $V(\widehat{m}_N(t)) = \int_t^\infty \left[ \widehat{m}_N'(t)\widehat{m}_N(t) + u^2(t) \right] dt$, where $\widehat{m}_N'(t)$ denotes the transpose of $\widehat{m}_N(t) \in \mathbb{R}^N$.

**Hierarchical policy learning.** We parameterize the value function $V_N$ by using an $N$-by-$N$ symmetric matrix $\theta_N$ in the way that $V_N(\widehat{m}_N; \theta_N) = \widehat{m}_N'\theta_N\widehat{m}_N$, and choose $\mathcal{L}$ to be the $L^2$-loss. Then, we vary the truncation order $N$ from 2 to 20, and the simulation results are shown in Figure 2. Specifically, Figure 2a shows the learned optimal polices $u_N^*(t)$ and optimal value functions (evaluated along the the system trajectories) $V(\widehat{m}_N^*(t); \theta_N^*)$ for all $N = 2, \ldots, 20$, from which we observe that both $u_N^*(t)$ and $V(\widehat{m}_N^*(t); \theta_N^*)$ converge to the corresponding shadowed regions as $N$ increases. To further verify the convergence behavior, we plot the projection error $\|V_N(\cdot; \theta_N^*) - V_{N-1}(\cdot; \theta_{N-1}^*)\| = \left( \int_0^\infty |V_N(\hat{m}_N^*(t); \theta_N^*) - V_{N-1}(\hat{m}_{N-1}^*(t); \theta_{N-1}^*)|^2 dt \right)^{1/2}$ and $\|u_N^* - u_{N-1}^*\| = \left( \int_0^\infty |u_N^*(t) - u_{N-1}^*(t)|^2 dt \right)^{1/2}$ in Figure 2b, both of which converge to 0 as desired. To be more specific, the projection error decreases very fast and starts to maintain a small value from the $N = 10$, which in turn demonstrates the high efficiency of the proposed algorithm.

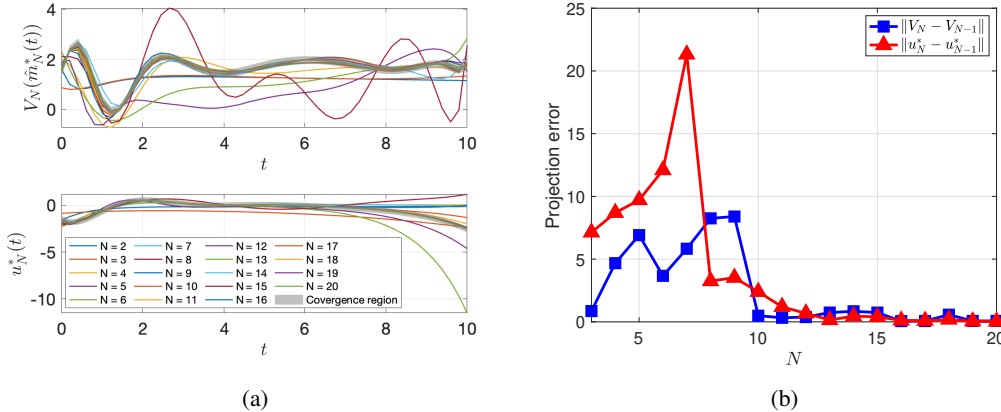

(a)         (b)

Figure 2: Reinforcement learning for the group system $\frac{d}{dt}x(t,\beta) = \beta x(t,\beta) + u(t)$ defined on the Hilbert space $L^2([-1,1])$ by using Algorithm 1, with the truncation order $N$ in the moment parameterization ranging from 2 to 20. Specifically, (a) shows the learned value functions (top) along the trajectories of the truncated moment systems driven by the learned optimal control policies (bottom), and both of them are convergent into the shadowed regions. (b) shows that the $L^2$-error between the value functions, as well as the optimal control policies, for the order $N$ and $N-1$ truncation converges to 0.

Moreover, to check the effectiveness of the moment kernel representation, we also apply the learned polices to the original group system, and compare the obtained trajectories, denoted by $x_N(t,\cdot)$, with those obtain by dekernelizing the optimal moment trajectories as $\widehat{x}_N(t,\cdot) = \sum_{k=0}^{N} \widehat{m}_N^*(t)\Phi_k(\cdot)$, with the simulation results shown in Figure 3. In particular, Figure 3a shows the maximum $L^2$-truncation error $\sup_t \left( \int_{-1}^{1} |x_N(t,\beta) - \widehat{x}_N(t,\beta)|^2 d\beta \right)^{1/2}$ with respect to the truncation order, which stabilizes to small values starting from $N = 10$, coinciding with the truncation order at which the truncation error stabilizes. More specifically, we also plot the time-evolution of the truncation error, that is, $\left( \int_{-1}^{1} |x_N(t,\beta) - \widehat{x}_N(t,\beta)|^2 d\beta \right)^{1/2}$ with respect to $t$, for each $N = 2, \ldots, 20$ in Figure 3b. Although the truncation error accumulates over time, it maintains a small value, and with the truncation order $N$ increasing, we observe a decrease in the increasing rate of the truncation error (the curve becomes flat). An efficient way to avoid the accumulation of the truncation error over time is to repeat restarting the reinforcement learning process after a short period of time.

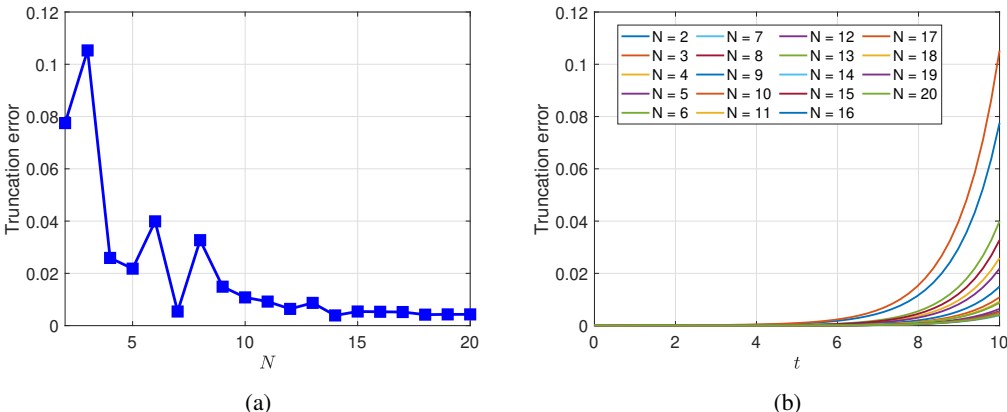

(a)         (b)

Figure 3: Analysis of the moment truncation error. In particular, (a) shows the time-evolution of the $L^2$-error between the trajectory of the ensemble system and the one recovered from the order-20 truncated moment system, both of which are driven by the optimal control policy shown in Figure 2a (top) for $N = 20$, and (b) plots the maximal truncation error with respect to the truncation order.

**Comparison with conventional reinforcement learning approach.** To further demonstrate the advantage of the proposed algorithm, we compare the above results with those obtained by directly applying an RL algorithm to the agents sampled from the group system. Following similar notations as above, here we use $N$ to denote the number of agents uniformly sampled from the the group system $\frac{d}{dt}x(t,\beta) = \beta x(t,\beta) + u(t)$ with $\beta \in [-1,1]$. These agents compose of a linear system defined on $\mathbb{R}^N$ of the form $\frac{d}{dt}x_N(t) = A_N x_N(t) + b_N u(t)$, in which $A_N \in \mathbb{R}^{N \times N}$ is the diagonal matrix with the entries $\beta_k = -1 + 2k/(N-1)$ for $k = 0, 1, \ldots, N-1$ on the diagonal and $b_N \in \mathbb{R}^{N \times 1}$ is the vector with every entry equal to 1. Correspondingly, the value function of the sampled system can be represented in the form of a Riemann sum as $V_N(x_N(t)) = \frac{2}{N-1}\sum_{k=0}^{N-1}\int_t^\infty e^{-2.5t}\big[x^2(t,\beta_i) + u^2(t)\big]dt = \frac{2}{N-1}\int_t^\infty e^{-2.5t}\big[x_N'(t)x_N(t) + u^2(t)\big]dt$. We then apply the value iteration algorithm with the sample size ranging from 2 to 20, and the simulation results are shown in Figure 4. Although, as shown in Figure 4a, the learned optimal value functions and policies have the trend to become closer as the sample size $N$ increases, their projection error definitely do not decrease to 0, and hence both of them fail to converge up to the sample size 20. As a comparison, we would like to reiterate that the proposed hierarchical learning algorithm for the moment kernelized group system converges at the truncation order 10 as shown in Figure 2b. However, the critical issue here is that the optimal value function starts to blow-up as we can observe from the top panel in Figure 4a, which is indeed the ill-pose problem caused by sampling the group system.

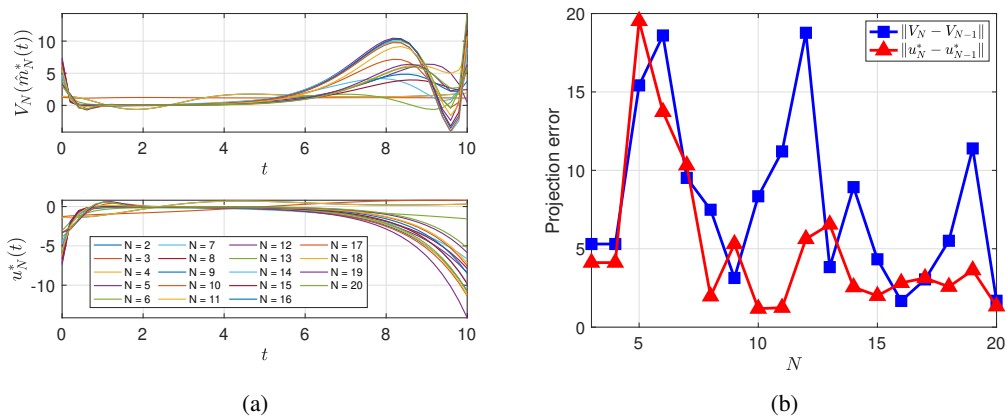

(a)  (b)

Figure 4: Reinforcement learning of sampled group systems by using the value iteration algorithm with the sample size $N$ ranging from 2 to 20. In particular, (a) shows the learned optimal value functions evaluated along the optimal trajectories (top) and the optimal policies (bottom) for all $N = 2, \ldots, 20$, and (b) plots the projection error for both the learned value functions and policies with respect to $N$.

## 5 CONCLUSIONS

In this paper, we propose a hierarchical algorithm for reinforcement learning of group systems consisting of large-scale, in the limit a continuum spectrum of, intelligent agents. In particular, we rigorously formulate such a reinforcement learning problem over an infinite-dimensional function space and then develop a moment kernel representation to transform the group system and its value function to the moment coordinates. By using the hierarchical structure induced by the "discrete nature" of the moment representation, that is, the moment coordinates always contain countably many components, we develop a reinforcement learning algorithm to learn the optimal policy of the group system hierarchically. In particular, each hierarchy in the proposed algorithm preserves the optimality of all the lower hierarchies, and this observation leads to the fast convergence of the algorithm, which is verified both theoretically and numerically. **Limitations.** Because this work focuses on the "continuous setting" with moments defined in terms of integrals, in practice, a sufficient amount of "discrete" data is required to obtain an accurate approximation of moments.

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
