## A  PROOF OF PROPOSITION 1

### A.1  EXISTENCE OF OPTIMAL POLICY

Let $J : \mathcal{U} \to \mathbb{R}$ be the objective function defined as $J(u) = V(x(0, \cdot))$, then it suffices to show that $J$ is a continuous function and the space of control policies $\mathcal{U}$ is compact. Then, by the general fact in analysis that any continuous function defined on a compact space has a minimum (Rudin, 1976), the optimal control policy exists.

**Continuity of the objective functional.**  Pick a sequence $\{u_k\}_{k \in \mathbb{N}}$ in $\mathcal{U}$ such that $u_k$ converges pointwisely to $u \in \mathcal{U}$, then we need to show $J(u_k) \to J(u)$. Let $x_k(t, \beta)$ be the trajectory (solution) of the ensemble system ensemble system in (1) driven by the control input $u_k(t)$, then $x_k(t, \beta)$ satisfies the fixed point equation (Arnold, 1978),

$$x_k(t, \beta) = x(0, \beta) + \int_0^t F(t, \beta, x_k(t, \beta), u_k(t))dt. \tag{5}$$

Similarly, let $x(t, \beta)$ be the trajectory of the ensemble system driven by the liming control function $u(t)$, then $x(t, \beta)$ satisfies the same equation as

$$x(t, \beta) = x(0, \beta) + \int_0^t F(t, \beta, x(t, \beta), u(t))dt \tag{6}$$

Taking the limit as $k \to \infty$ for both sides of the equation in (5) yields

$$\lim_{k \to \infty} x_k(t, \beta) = x(0, \beta) + \lim_{k \to \infty} \int_0^t F(t, \beta, x_k(t, \beta), u(t))dt$$

$$= x(0, \beta) + \int_0^t \lim_{k \to \infty} F(t, \beta, x_k(t, \beta), u_k(t))dt$$

$$= x(0, \beta) + \int_0^t F(t, \beta, \lim_{k \to \infty} x_k(t, \beta), \lim_{k \to \infty} u_k(t))dt$$

$$= x(0, \beta) + \int_0^t F(t, \beta, \lim_{k \to \infty} x_k(t, \beta), u(t))dt$$

where the second and third equalitites follow from the dominant convergence theorem and continuity of $F$, respectively (Folland, 2013). Because the solution of the ensmeble system in (1), equivalently, the fixed point equation in (6), is unique for each $\beta \in \Omega$ by Assumptions S2, we conclude that $x(t, \beta) = \lim_{k \to \infty} x_k(t, \beta)$ for all $\beta \in \Omega$. Applying the dominant convergence theorem again to $J$ with the continuity of $r$ and $K$, we also obtain

$$\lim_{k \to \infty} J(u_k) = \lim_{k \to \infty} \int_\Omega \Big[ \int_0^T r(x_k(t, \beta), u_k(t))dt + K(x_k(T, \beta)) \Big] d\beta$$

$$= \int_\Omega \Big[ \int_0^T r(\lim_{k \to \infty} x_k(t, \beta), \lim_{k \to \infty} u_k(t))dt + K(\lim_{k \to \infty} x_k(T, \beta)) \Big] d\beta$$

$$= \int_\Omega \Big[ \int_0^T r(x(t, \beta), u(t))dt + K(x(T, \beta)) \Big] d\beta = J(u),$$

indicating the continuity of $F$ as desired.

**Compactness of the space of control policies.**  By Assumption S1 that control inputs in $\mathcal{U}$ are bounded by a constant $A$ uniformly, $\mathcal{U}$ is the closed ball with the radius $A$ centered at the 0 control input in the space of all bounded functions from $[0, T]$ to $\mathbb{R}^m$. Then, by the Alaoglu's Theorem (Folland, 2013), $\mathcal{U}$ is compact in the weak* topology, which coincides with the topology of pointwise convergence as used in the proof of the continuity of $J$ above, concluding the proof.

### A.2  REGULARITY OF VALUE FUNCTION

In particular, we would like to show that the value function $V$ of the infinite-time horizon ensemble reinforcement learning problem is bounded. Moreover, if $\lambda > L$, the Lipschitz constant of $\bar{F}$, then $V$

is Lipschitz continuous; if $0 < \lambda \leq L$, then $V$ is Hölder continuous for some exponent $0 < \alpha < 1$. In addition, owing to the one-to-one correspondence between ensemble states and the associated moment sequences, the proof can be equivalently carried out by using the moment coordinates.

The boundedness of the value function $V$ directly follows from that of the reward function and integrability of the discount factor as

$$|J(u)| \leq \int_0^\infty e^{-\lambda t}|r(m(t), u(t))|dt \leq \max_{m \in \mathcal{M}, a \in [-A, A]} |r(m, a)| \cdot \int_0^\infty e^{-\lambda t}dt$$

$$= \frac{1}{\lambda} \max_{m \in \mathcal{M}, a \in [-A, A]} |r(m, a)| < \infty.$$

To show the Hölder continuity of $V$, pick $m_0, m_0' \in \mathcal{M}$, by the definition of $V$, for any $\varepsilon > 0$, there is some $u \in \mathcal{U}$ such that

$$V(\bar{m}) + \varepsilon \geq \int_0^\infty e^{-\lambda t}r(\bar{m}(t), u(t))dt$$

with $\bar{m}(t)$ satisfying the system $\frac{d}{dt}\bar{m}(t) = \bar{F}(\bar{m}(t), u(t))$ with $\bar{m}(0) = \bar{m}_0$. Let $m(t)$ be the trajectory of the system driven by the same control input but with a different initial condition $m(0) = m_0$, then we have

$$V(m_0) - V(\bar{m}_0) \leq \int_0^\infty e^{-\lambda t}\big(r(m(t), u(t)) - r(\bar{m}(t), u(t))\big)dt + \varepsilon$$

$$\leq \int_0^\infty e^{-\lambda t}C|m(t) - \bar{m}(t)|dt + \varepsilon,$$

where we use the Lipschitz continuity of $r$. Interchanging the role of $m(t)$ and $\bar{m}(t)$ yields $V(\bar{m}_0) - V(m) \leq \int_0^\infty e^{-\lambda t}C|\bar{m}(t) - m(t)|dt + \varepsilon$. Together that $\varepsilon$ is arbitrary, we obtain

$$|V(m) - V(\bar{m})| \leq C\int_0^\infty e^{-\lambda t}|m(t) - \bar{m}(t)|dt. \tag{7}$$

To estimate the term $|m(t) - \bar{m}(t)|$, we notice

$$\frac{d}{dt}\big(m(t) - \bar{m}(t)\big) = \bar{F}(m(t), u(t)) - \bar{F}(\bar{m}(t), u(t)) \leq L|m(t) - \bar{m}(t)|.$$

Similarly, interchanging the role of $m(t)$ and $\bar{m}(t)$ yields

$$\frac{d}{dt}|m(t) - m'(t)| \leq L|m(t) - \bar{m}(t)|,$$

which is well-defined for all $t$ since $m(t) - \bar{m}(t) \neq 0$, otherwise, it will violate the uniqueness of solutions for the ordinary differential equation $\frac{d}{dt}m(t) = \bar{F}(m(t), u(t))$. Now, applying Gronwall's inequality gives

$$|m(t) - \bar{m}(t)| \leq e^{Lt}|m_0 - \bar{m}_0|.$$

Consequently, we have

$$|V(m) - V(\bar{m})| \leq C|m - \bar{m}_0| \int_0^\infty e^{-(\lambda - L)t}dt.$$

Therefore, if $\lambda > L$, then

$$|V(m_0) - V(\bar{m}_0)| \leq \frac{C}{\lambda - L}|m_0 - \bar{m}_0|$$

holds so that $V$ is Lipschitz continuous.

Next, if $0 < \lambda \leq L$, we pick $0 < \alpha < 1$ such that $\lambda > \alpha L$. Note that because $r(\cdot, u)$ is Lipschitz continuous by Assumption C2, it is also $\alpha$-Hölder continous, i.e., $|r(m(t), u(t)) - r(\bar{m}(t), u(t))| \leq C|m(t) - \bar{m}(t)|^\alpha$, which gives a variant of (7) as

$$|V(m_0) - V(\bar{m}_0)| \leq C\int_0^\infty e^{-\lambda t}|m(t) - \bar{m}(t)|^\alpha dt.$$

Together with $|m(t) - \bar{m}(t)|^\alpha \leq e^{\alpha L t}|m - \bar{m}|^\alpha$, we have

$$|V(m_0) - V(\bar{m}_0)| \leq C|m_0 - \bar{m}_0|^\alpha \int_0^\infty e^{-(\lambda - \alpha L)t}dt \leq \frac{C}{\lambda - \alpha L}|m_0 - \bar{m}_0|^\alpha,$$

giving Hölder continuouity of $V$ with Hölder exponent $\alpha$.

## B   MOMENT CONVERGENCE OF TRUNCATED REINFORCEMENT LEARNING PROBLEMS

At first, as proved in Appendix A, the space of control polices $\mathcal{U}$ is compact, and hence the sequence $\{u_N^*\}_{N\in\mathbb{N}}$ has a convergent subsequence $\{u_{N_i}^*\}_{i\in\mathbb{N}}$ and we denote the limit by $u^*$. It then remains to show that $V_{N_i} \to V$ as $i \to \infty$ and $u^*$ solves the Hamilton-Jacobi-Bellman equation along the trajectory $m^*(t)$ of the moment system steered by $u^*(t)$ as

$$\frac{\partial V}{\partial t} + DV(t, m^*(t)) \cdot \bar{F}(t, m^*(t), u^*(t)) + \bar{r}(m^*(t), u^*(t)) = 0, \quad V(T, m(T)) = \bar{K}(m(T)). \tag{8}$$

To this end, let $V_{N_i}$ denote the value function for the order $N$ truncated ensemble reinforcement learning problem, then for all $0 \le t \le T$, $V_{N_i}(t, \cdot)$ is essentially the restriction of $V(t, \cdot)$ to the space consisting of the order $N_i$ truncated moment sequences $\hat{m}_{N_i}$, which implies $V(t, \hat{m}_N) = V_{N_i}(t, \hat{m}_{N_i})$. The Lipschitz continuity of $V$ then yields

$$|V_{N_i}(t, \hat{m}_{N_i}) - V_{N_i}(t', \hat{m}'_{N_i})| = |V(t, \hat{m}_{N_i}) - V(t', \hat{m}'_{N_i})| \le C\big(|t - t'| + |\hat{m}_{N_i} - \hat{m}'_{N_i}|\big)$$

for any time $0 \le t, t' \le T$ and order $N_i$ truncated moment sequences $\hat{m}_{N_i}$ and $\hat{m}'_{N_i}$. This implies the family of values functions $\{V_{N_i}\}_{i\in\mathbb{N}}$ are Lipschitz continuous with the same Lipschitz constant. By the definition, it immediately follows that $\{V_{N_i}\}_{i\in\mathbb{N}}$ is uniformly equicontious (Rudin, 1976). Together with the boundedness of $V$, and hence all $V_{N_i}$, we conclude that $V_{N_i}$, maybe by passing to a subsequence, converges uniformly to a function $V'$ on compact sets by Arzela-Ascoli Theorem (Folland, 2013). As a consequence $V'$ is also continuous, since each $V_{N_i}$ is. At last, we need to show the $V'$ satisfies the Hamilton-Jacobi-Bellman equation in (8).

We first note that because $u_{N_i}^*$ is the optimal control policy, it necessarily satisfies

$$\frac{\partial V_{N_i}}{\partial t}(t, \hat{m}_{N_i}^*(t)) + DV_{N_i}(t, \hat{m}_{N_i}^*(t)) \cdot \hat{F}_{N_i}(t, \hat{m}_{N_i}^*(t), u_{N_i}^*(t)) = 0,$$
$$V_{N_i}(T, \hat{m}_{N_i}^*(T)) = \hat{K}_{N_i}(\hat{m}_{N_i}^*(T)), \tag{9}$$

where $\hat{m}_{N_i}^*(t)$ is the corresponding optimal trajectory and $\hat{K}_{N_i}$ is the restriction of $K$ to the space of order $N_i$ truncated moment sequences. In addition, as the solution of the truncated moment system $\frac{d}{dt}\hat{m}_{N_i}^*(t) = \hat{F}_N(t, \hat{m}_{N_i}^*(t), \hat{u}_{N_i}^*(t))$, $\hat{m}_{N_i}^*(t)$ satisfies the fixed point equation $\hat{m}_{N_i}^*(t) = \hat{m}_{N_i}^*(0) + \int_0^t \hat{F}_N(s, \hat{m}_{N_i}^*(s), \hat{u}_{N_i}^*(s))ds$. The continuity of $\hat{F}_N$ and the dominant convergence theorem together imply

$$\lim_{i\to\infty} \hat{m}_{N_i}^*(t) = \lim_{i\to\infty} \hat{m}_{N_i}^*(0) + \lim_{i\to\infty} \int_0^t \hat{F}_N(s, \hat{m}_{N_i}^*(s), \hat{u}_{N_i}^*(s))ds$$
$$= m^*(0) + \int_0^t \hat{F}_{N_i}(s, \lim_{i\to\infty} \hat{m}_{N_i}^*(s), \lim_{i\to\infty} \hat{u}_{N_i}^*(s))ds$$
$$= m^*(0) + \int_0^t F(s, \lim_{i\to\infty} \hat{m}_{N_i}^*(s), u^*(s))ds,$$

which reveals the convergence of $\hat{m}_{N_i}^*(t)$ to a trajectory $m^*(t)$ solving the untruncated moment system $\frac{d}{dt}m^*(t) = F(t, m^*(t), u^*(t))$. More importantly, the convergence of $\hat{m}_{N_i}^*(t)$ also implies that all these trajectories lay in a compact space, and hence the convergence of $V_{N_i}$ on compact sets applies to the Hamilton-Jacobi-Bellman equation in (9) as

$$0 = \lim_{i\to\infty} \frac{\partial V_{N_i}}{\partial t}(t, \hat{m}_{N_i}^*(t)) + \lim_{i\to\infty} DV_{N_i}(t, \hat{m}_{N_i}^*(t)) \cdot \hat{F}_{N_i}(t, \hat{m}_{N_i}^*(t), u_{N_i}^*(t))$$
$$= \frac{\partial}{\partial t} \lim_{i\to\infty} V_{N_i}(t, \hat{m}_{N_i}^*(t)) + D\big[\lim_{i\to\infty}(V_{N_i}(t, \hat{m}_{N_i}^*(t)))\big] \cdot \lim_{i\to\infty} \hat{F}_{N_i}(t, \hat{m}_{N_i}^*(t), u_{N_i}^*(t))$$
$$= \frac{\partial}{\partial t} V'(t, \hat{m}_{N_i}^*(t)) + DV'(t, \lim_{i\to\infty} \hat{m}_{N_i}^*(t)) \cdot \bar{F}(t, \lim_{i\to\infty} m_{N_i}^*(t), \lim_{i\to\infty} u_{N_i}^*(t))$$
$$= \frac{\partial}{\partial t} V'(t, m^*(t)) + DV'(t, m^*(t)) \cdot \bar{F}(t, m^*(t), u^*(t))$$

together with the terminal condition

$$0 = \lim_{i \to \infty} V_{N_i}(T, \hat{m}^*_{N_i}(T)) - \lim_{i \to \infty} \hat{K}_{N_i}(\hat{m}^*_{N_i}(T)) = V'(T, \lim_{i \to \infty} \hat{m}^*_{N_i}(T)) - \bar{K}(\lim_{i \to \infty} \hat{m}^*_{N_i}(T))$$
$$= V'(m^*(t)) - \bar{K}(m^*(T)),$$

where the changes of limits and differentiations follo from the equicontinuity of the sequence of functions $\{V_{N_i}\}$ (Rudin, 1976). This shows that $V'$ satisfies the Hamilton-Jacobi-Bellman equation, and hence $V = V'$ holds by the uniqueness of the (viscosity) solution of the Hamilton-Jacobi-Bellman equation (Evans, 2010). As a consequence, $u^*(t)$ is the optimal control policy and $m^*(t)$ is the optimal moment trajectory.