# OpenReview forum: "Reinforcement Learning for Large Group Systems using Hierarchical Kernel Representations"
_ICLR.cc/2024/Conference — Submitted to ICLR 2024_

### Official Review · Reviewer_VDh3 · 2023-10-26

**Soundness:** 2 fair
**Presentation:** 2 fair
**Contribution:** 1 poor
**Rating:** 3
**Confidence:** 3

**Summary:**

This paper considers "reinforcement learning" (in fact, the *planning* problem) in a special setting of large-scale multi-agent dynamical systems. The learning objective is to find a universal policy $u(t)$ for a massive number of agents, where each agent is characterized by a type parameter $\beta \in \varOmega$ that follows a known distribution, and the dynamics of an agent with type may be dependent on its type $\beta$. To solve the proposed problem, the paper proposes a hierarchical algorithm that leverages a novel moment kernel representation for group systems, with convergence guarantee and detailed simulation results in a specific LQR system.

**Strengths:**

The paper is written in an understandable way. It attempts to formulate a novel sequential decision-making problem in a large-scale setting different from MDPs/Markov games. The mathematical formulation and proofs appear to be correct.

**Weaknesses:**

The reviewer believes that this paper needs a significant amount of further work to make it publishable. Specifically, the reviewer has the following concerns:
1. What is the motivation for considering such a special setting?
    * It's hard to consider a setting where there are a massive number of agents, each follows a (potentially different) *type-dependent dynamics*, yet all of them share a common policy $u(t)$ that is *type-independent*.
    * The paper fails to sufficiently motivate the new setting. Though it seems that the authors are willing to provide an example in Section 2.2, the link is dead and nor can the reviewer find it in the appendix.
    * The agent type follows a uniform distribution over $\varOmega$, which hardly makes any sense. For large-scale systems at least a known distribution $\mu$ over $\varOmega$ should be considered. Will this make the results significantly different (better or worse)?
2. Why is this referred to as RL?
    * Terminology-wise, RL specifically refers to the learning-based solutions to MDPs or Markov games. Not all sequential decision-making problems with rewards fall into the category of RL.
    * There is no learning involved in the proposed setting. All components are known, including the system dynamics, the agent types (in this case, the distribution of types) and rewards. The formulation can be regarded as a planning problem at best.
    * To qualify the algorithm as "learning", at least some component should be initially unknown, and inferred from data collected along the trajectory.
3. Why is the hierarchical moment-kernel-based algorithm necessary?
    * The introduction of the moment kernel technique is not motivated. What's the fundamental challenge of the problem? Why the moment kernel technique helps to settle it? Now it appears to the reviewer that the authors just randomly select a fancy approach.
    * Even for the moment kernel, the usage of Chebyshev polynomials in the exemplary setting is not motivated as well. Can other moment kernels be used in place of it? How does it impact the performance?
    * It is unclear why the algorithm needs to be hierarchical. Is it a simple binary-lifting technique to find a truncation that yields an error below threshold? It is even not clear which part of the algorithm is "hierarchical", and in what sense.
4. Why bother to include a detailed analysis of a specific system?
    * It is not convincing enough to show the correctness and performance of the proposed algorithm via an analysis of *one* specific artificially-designed system.
    * If the paper is application-oriented, please include the modelling of a real-world use case and show the end-to-end performance.
    * If the paper is theory-oriented, please include theoretical guarantees of the algorithm under mild assumptions (e.g., regret bounds, sample complexity, etc.).
5. The paper does not include sufficient literature review and discussion about the results to let the reviewer see the significance of the results.

**Questions:**

Please refer to the weaknesses part above.

---

### Official Review · Reviewer_tFnE · 2023-10-30

**Soundness:** 3 good
**Presentation:** 2 fair
**Contribution:** 2 fair
**Rating:** 5
**Confidence:** 4

**Summary:**

This paper introduces a framework for reinforcement learning (RL) in the context of coordinating (super) large populations of intelligent agents, referred to as group systems. The main contributions of the paper are as follows:

1. The paper formulates populations of intelligent agents as deterministic group systems defined on function spaces. This approach allows for RL problems to be framed in a functional setting, making it applicable to massive-scale populations of agents, regardless of their size.

2. The authors use a moment kernel representation for group systems. This representation provides a new way to understand and analyze the behavior of these populations, enhancing the modeling and control of large agent populations.

3. The paper presents a hierarchical RL algorithm for group systems. The hierarchy here is different from, say, goal-based reinforcement learning settings. The moment sequence can be truncated at a given order, and the truncated sequences are called "hierarchy". This is more like an approximation algorithm.

In summary, the paper addresses the challenge of coordinating massive populations of intelligent agents and offers a comprehensive framework for RL in group systems, including a moment kernel representation and an approximation algorithm.

**Strengths:**

The paper demonstrates strength in various dimensions:

1. It introduces a novel (as far as the reviewer is concerned) approach for coordinating large populations of intelligent agents, departing from traditional RL.

2. The paper effectively communicates complex concepts, although some parts could benefit from further clarification.

**Weaknesses:**

1. (Major) While the paper presents an intriguing framework for reinforcement learning in the context of coordinating large populations of intelligent agents, several aspects related to its applicability warrant critical examination. The reviewer has concerns about the representational power of the moment kernel introduced.  Its limitations and practical applicability to real-world scenarios remain unclear. Furthermore, the focus of the experimental evaluation is on Linear Quadratic Regulators (LQR). While LQR is undoubtedly a valuable benchmark problem in control theory, it does not reflect the diverse range of applications for which this framework is purportedly designed. The absence of experiments on scenarios and the limited scope of the study on LQR raise questions about the broader applicability and versatility of the proposed hierarchical algorithm.

1.1 Can the authors show an example where the motivating examples presented in the introduction, such as neural stimulation and quantum control, are represented by the moment kernel?

1.2 Can the authors provide experimental results on more scenarios?

2. Some related works are worth discussing, like [1].

3. About baselines. Why the authors choose value iteration as the baseline RL algorithm? How value iteration is carried out on a continuous state space? There are more powerful multi-agent reinforcement learning algorithms designed for large populations (include but not limited to mean-field methods). Comparing with these algorithms would significantly improve the soundness of the proposed method.

4. (Major) Please help the reviewer understand why a hierarchical framework is necessary. Why can't we start directly from a long, truncated sequence? What are the advantages of beginning with a relatively short sequence?

### Minor problems
1. Where is the example demonstrating that it is generally impossible to simultaneously minimize the (discounted) future rewards for all the agents by a universal policy?

2. In the paragraph below equation (2), what does the notation $u_{[t,\infty)]}$ mean?

[1] Zheng, L., Yang, J., Cai, H., Zhou, M., Zhang, W., Wang, J. and Yu, Y., 2018, April. Magent: A many-agent reinforcement learning platform for artificial collective intelligence. AAAI 2018 (Vol. 32, No. 1).

**Questions:**

The reviewer has concerns about the applicability of the moment kernel, the soundness of experimental studies, and the intuitive idea of the hierarchical learning method. Please answer the questions in the weakness section.

---

### Official Review · Reviewer_ns7v · 2023-11-03

**Soundness:** 3 good
**Presentation:** 3 good
**Contribution:** 3 good
**Rating:** 6
**Confidence:** 2

**Summary:**

This paper proposes a framework for representing the deterministic multi-agent system with a large population as a group system. Based on the framework, a moment kernel representation of the system and a hierarchical RL algorithm algorithm are proposed to solve the system. The development of the framework and the solution is supported by theoretical proof. The experiment results indicate better convergence of the proposed method compared to conventional value iteration.

**Strengths:**

The paper is well written and the contribution is novel. The statements in the paper are well supported by theoretical proof and empirical results.

**Weaknesses:**

It is a bit difficult for me to understand how generalizable the proposed method is on real-world problems. An experiment with more realistic settings or just an explanation of the relation of the experiments in the paper with real-world problems would be great.

As mentioned in the paper, the amount of data affetcs the accuracy of the approximation of moments. It would be good to have some analysis of the performance of the method and the amount of the data.

**Questions:**

Can authors provide:
1. An experiment with more realistic settings or an explanation of the relation of the experiments in the paper with real-world problems? This would help readers to understand how generalizable the proposed method is.

2. Some analysis of the performance of the method and the amount of the data.

---

### Meta-Review · Area_Chair_gt8T · 2023-12-09

**Metareview:**

Paper introduces a novel approach for controller a cohort of intelligent agents by leveraging Hierarchical Kernel Representations in RL methods. Paper is decently written.

However, reviewers have concerns on motivation of the problem, validity of the LQR baselines in experiments, more SOTA comparisons, and the representation power of the kernal representation proposed and its induced approximation error analysis in corresponding RL algorithms. More experiments, detailed literature review, general analysis and motivation of the hierarchical representation are also requested by reviewers. Therefore, paper cannot be accepted at this point before aforementioned issues are addressed.

**Justification For Why Not Higher Score:**

Reviewer pointed at several important issues of the paper and they need to be addressed before paper acceptance.

**Justification For Why Not Lower Score:**

N/A

---

### Decision · Program_Chairs · 2024-01-16

Reject